# The Comparison of the Environmental Impact of Waste Mineral Wool and Mineral in Wool-Based Geopolymer

**DOI:** 10.3390/ma15062050

**Published:** 2022-03-10

**Authors:** Beata Łaźniewska-Piekarczyk, Monika Czop, Dominik Smyczek

**Affiliations:** 1Department of Building Processes and Building Physics, Faculty of Civil Engineering, The Silesian University of Technology, Akademicka 5, 44-100 Gliwice, Poland; dominik.smyczek@saint-gobain.com; 2Department of Technologies and Installations for Waste Management, Faculty of Energy and Environmental Engineering, The Silesian University of Technology, Konarskiego 18, 44-100 Gliwice, Poland; monika.czop@polsl.pl; 3Saint Gobain Construction Products Polska Sp z o.o., ul. Okrężna 16, 44-100 Gliwice, Poland

**Keywords:** production waste, rock wool, glass wool, loss on ignition, pollutants washout, geopolymer

## Abstract

Waste generated in fine wool production is homogeneous and without contamination, which increases its chances of reuse. Waste mineral wool from demolition sites belongs to the specific group of waste. However, the storage and collection require implementing restrictive conditions, such as improper storage of mineral wool, which is highly hazardous for the environment. The study focuses on the leachability of selected pollutants (pH, Cl^−^, SO_4_^2−^) and heavy metals (Ba, Co, Cr, Cu, Ni, Pb, Zn) from the waste mineral wool. As a solution to the problem of storing mineral wool waste, it was proposed to process it into wool-based geopolymer. The geopolymer, based on mineral wool, was also assessed regarding the leaching of selected impurities. Rock mineral wool is very good for geopolymerisation, but the glass wool needs to be completed with additional components rich in Al_2_O_3_. The research involved geopolymer prepared from mineral glass wool with bauxite and Al_2_O_3_. So far, glass wool with the mentioned additives has not been tested. An essential aspect of the article is checking the influence of wool-based geopolymer on the environment. To investigate the environmental effects of the wool-based monolith and crushed wool geopolymers were compared. Such research has not been conducted so far. For this purpose, water extracts from fragmented geopolymers were made, and tests were carried out following EN 12457-4. There is no information in the literature on the influence of geopolymer on the environment, which is an essential aspect of its possible use. The research results proved that the geopolymer made on the base of mineral wool meets the environmental requirements, except for the pH value. As mentioned in the article, the geopolymerisation process requires the dissolution of the starting material in a high pH (alkaline) solution. On the other hand, the pH minimum 11.2 value of fresh geopolymer binder is required to start geopolymerisation. Moreover, research results analysed in the literature showed that the optimum NaOH concentration is 8 M. for the highest compressive strength of geopolymer. Therefore, the geopolymer strength decreases with NaO concentration in the NaOH solution. Geopolymers glass wool-based mortars with Al_2_O_3_ obtained an average compressive strength of 59, the geopolymer with bauxite achieved about 51 MPa. Thus, Al_2_O_3_ is a better additional glass wool-based geopolymer than bauxite. The average compressive strength of rock wool-based geopolymer mortar was about 62 MPa. The average compressive strength of wool-based geopolymer binder was about 20–25 MPa. It was observed that samples of geopolymers grout without aggregate participation are characterised by cracking and deformation.

## 1. Introduction

Proper waste management should be directed on priority circular economy. The main principle of circular economy is to maintain the value of raw materials, ready-made goods, and finished products if possible while minimising waste. The generated waste, however, should be reused, i.e., recycled. In 2020, 123 million tons of waste was generated, including 109.47 thousand tons from various economic activity branches; as for the construction industry, 7.3 million tons was generated in 2020. It is an increase of 60% compared to 2019 [1].

Mineral wool is a popular construction material, so it often functions in waste management obtained at the construction site. The producer is responsible for the entire waste management process (usually the contractor on the construction site) and becomes the owner of the waste at the time of its production. In addition, he is responsible for the costs associated with managing and storing the waste [2].

According to the applicable regulations, construction waste management should be based on the waste management hierarchy. At the very beginning, the institution causing waste production should prevent its generation at every production stage (in the case of mineral wool, it is a factory producing it for the construction of buildings). In the next step, the waste owner should be prepared to reuse it, applying the principle of proximity (process them at the point of origin). In practice, many waste materials are reused at the construction site to save construction costs. However, it should be noted that the recycled material to be reused for construction purposes needs to undergo control tests determining its mechanical and physical properties. Ultimately, it all comes down to recycling, which is currently crucial to waste management. Unfortunately, wool as waste is generally put into storage. For the environment, it is the least favourable solution. It occupies space in landfills, thus leaving less and less room for people and all other creatures inhabiting the Earth.

As part of the search for a replacement, fewer emission binders for concrete production developed geopolymers in construction. However, the practical application of these materials is still minimal. The production of every ton of cement puts one tonne of CO_2_ into the atmosphere [1,3,4]. According to various estimates, the synthesis of geopolymers absorbs 2–3 times less energy than Portland cement and causes liberation of 4–8 times less CO_2_ [1,5,6,7].

The first geopolymer applications can be dated to the beginning of the 1970s. At that time, fire-resistant chipboards were developed with a wooden core covered with two geopolymer coatings. In 1978–1980, a metakaolin-based liquid geopolymer binder containing a soluble alkali silicate was developed at the Cordis-Laboratory. It was the first mineral resin produced [8,9]. Rocla has developed a technology for producing sewage pipes from steel-reinforced geopolymers [9]. They have successfully produced pipes with diameters ranging from 37.5 cm to 1.8 mm and adapted the conventional concrete prestressing technology to make railway sleepers from GPC prestressed geopolymer concrete. Geopolymer sleepers have been alternated with conventional sleepers on the main track line since 2002, and no problems were identified [10]. Currently, low-profile sleepers made of GPC are being developed as an alternative to wooden sleepers. The Czech Republic has developed a new inorganic binder system for self-hardening masses based on a geopolymer binder, a viscous liquid with a low degree of polymerisation used to prepare self-hardening masses. As a result of the hardener, polymerisation increases and a polymer with a high binding capacity is formed. These materials enable foundries to produce cores from self-hardening materials ecologically and economically [11]. Attempts have also been made to use geopolymer materials in drilling, mining, and hydro-technical construction for strengthening and sealing [12,13]. Alkaline activated geopolymer concrete has been commercialised in Australia under E-Crete (TM) name and is very popular among customers. E-Crete (TM) concrete reduced greenhouse gas emissions in concrete production by nearly 80% compared to concretes made of Portland cement [1]. Until now, many different types of surfaces have been made of geopolymers, such as pavements, playgrounds slabs, building foundations, and many others. Rocla was the first company globally to launch geopolymer concrete products on a commercial scale [3]. In Australia, at the University of Global Change Queensland (GCI), the world’s first public utility building was designed and constructed in 2013; the structural elements made of geopolymer concrete were used in building constructions.

Geopolymer mixtures have good fluidity, which allows for an accurate representation of the form and the possibility of obtaining surfaces with various degrees of smoothness [3,4,12,13,14,15,16,17,18,19,20]. The concrete produced on the base of the geopolymer binder does not undergo the phenomenon of shrinkage cracks [12,18,19]. Geopolymer concrete is characterised by high compressive strength, minimal shrinkage and negligible creep, better adhesion and high resistance to acid and sulphate corrosion [1,3,4,5,6,11,12,13,16,17,18,19,21,22,23,24,25,26,27,28,29]. Some researchers also found that this concrete is also resistant to carbonate corrosion and possesses a very high fire resistance [15] and a high resistance to UV radiation [30]. Therefore, it is possible to the printing of geopolymers [22]. Geopolymers have excellent pouring properties and perfectly match even complex forms. In addition to the applications mentioned above, they are also used to produce various types of small-sized products, such as decorative elements. They have also found their use as injections strengthening the soil. In addition, geopolymer concrete is used to immobilise hazardous waste, including radioactive waste. The availability of several geopolymer-based products will increase rapidly, mainly due to the properties of this material and the fact that an increasing number of enterprises and part of the society are beginning to become convinced of this type of product.

Geopolymers are inorganic polymers formed by combining aluminosilicate with an activator and water. A geopolymer recipe could be based on a variety of aluminosilicate materials. The geopolymer microstructure considers an aluminosilicate polymer, synthetically produced using silicon (Si) and aluminium (Al), geologically acquired from minerals. The chemical composition of the geopolymer is similar to the chemical composition of zeolite, but the structure of the geopolymer is different [27]. The process of geopolymerisation is a chemical reaction between an alkali solution and a source material containing aluminosilicate (FA). It gives a three-dimensional polymeric chain and ring structure consisting of Si–O–Al–O bonds, as reported by Figure 1 [30,31]: -PSDS Si-O-Al-O-Si-O-Si-O—poly(sialate-disiloxo),-PSS Si-O-Al-O-Si-O—poly(sialate-siloxo),-PS Si-O-Al-O—polysialate.

**Scheme 1 materials-15-02050-sch001:**
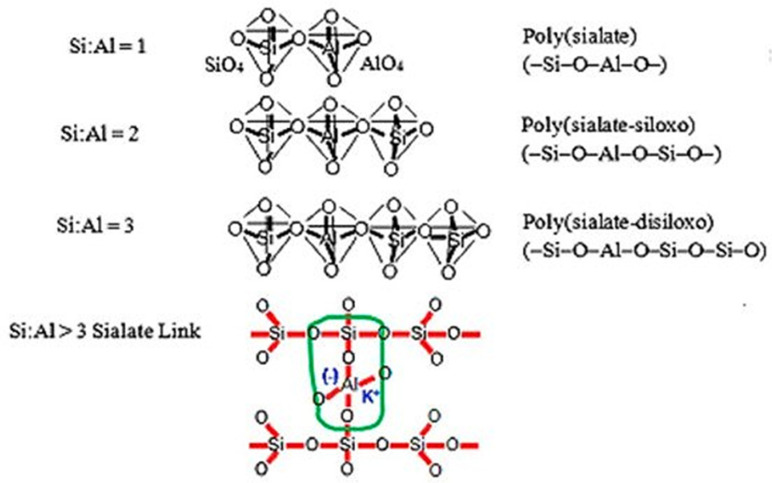
Structure of polysialates [29].

The reactions can occur at room temperature. Therefore, it can be considered energy and source efficient and much cleaner. Two main categories of the division of geopolymers are accepted. The first one considers the elementary units of polymeric chains [5,28,29,32].

The second classification of geopolymers concerns their origin and, more specifically, the type of pozzolanic aluminosilicate material. The oxides of silicon and aluminium constitute the bases of the geopolymer composition, and the additives of metal cations such as sodium or potassium constitute the stabilising material here.

Table 1 presents the chemical composition of the three most popular and most readily available materials from which geopolymeric cement can be produced. As it is evident, the contents of individual oxides are similar. All three are built on the base of silicon and aluminium oxides.

Due to this criterion, we distinguish geopolymers formed on the base of the following: fly ash, metakaolin, various types of rocks, volcanic agglomerates, silicas, fossil materials. Geopolymers are also from rock materials such as phosphorites [34] or pozzolanic volcanic ashes [35,36]. An important step forward was the use of geopolymer technology in waste management. Establishing the geopolymer recipe on industrial waste brings many environmental benefits and reduces costs related to waste management [3,4,12,13,14,17,18,19,20]. In producing geopolymers, such wastes as fly ash, diabase scrubbers [37] or bio-carbon resulting from biomass [38] are used.

From a chemical and mineralogical point of view, mineral rock wool is an ideal material for geopolymerisation (Table 2). 

However, in the case of glass wool, Al_2_O_3_ is too low, and it is necessary to add another waste or natural component rich in Al_2_O_3_. Moreover, the essential factors which affect the process of geopolymerisation are given below [34,35]:Type of raw materials containing aluminosilicateThe surface area of solid raw materialsGlassy phase content in the raw materialAmount of aluminium and reactive siliconPresence of iron, calcium, and inert particles in FACuring temperature and pressureDuration of curingType of curing (conventional heating or microwave heating)Type and concentration of alkaliesAlkaline liquid-to-raw material ratioH_2_O to Na_2_O molar ratioWater to Geopolymer solids ratioNa_2_O to SiO_2_ ratioSiO_2_ to Al_2_O_3_ ratio

The properties of geopolymers depend on the type of base material, the type and amount of activator used and their production technology, i.e., mixing time, material fragmentation, hardening temperature, humidity, hardening time and the amount of added water [3,4,9,12,13,14,17,18,19,20]. In addition, types of used alkaline activators, such as NaOH solution, KOH solution, a mixture of NaOH solution-water glass, and a mixture of KOH solution-water glass, are also essential to achieve a high compressive strength the geopolymers [15].

Moreover, there is the possibility of adjusting the porosity (it is possible to attain low porosity below 2.5% or high porosity above 20%, depending on needs, up to foaming geopolymers (as insulating materials) [33,37]. If the geopolymer is foamed in the production process, it might become lightweight and partly insulating [6,36]. On the other hand, in an almost pore-free form, the geopolymer may be used to protect walls against moisture and fungi formation [37,38,39]. 

Geopolymers are becoming essential in waste neutralisation technologies, especially in the neutralisation of hazardous waste [3,5,29,40]. Similarly, as material approved for use in construction, geopolymers should meet environmental regulations regarding the environment and human health safety.

The research involved geopolymer prepared from mineral rock and glass wool. The authors have already attempted to create a geopolymer from the mineral wool of the publication [6,7]. Depending on the production method, wool geopolymers may have different properties. For example, due to the high degree of grinding, the wools might have high strength as of 50 MPa [6] or, due to the addition of other components, they may have a porous structure which causes the geopolymer to develop different properties [6]. The rock wool has an adequate proportion of Si and Al to geopolymerisation, but glass wool has got to a low amount of Al. Therefore, the glass wool needs an additional component rich in Al_2_O_3_. As a source of Al, the bauxite was used. So far, glass wool with the mentioned additives has not been tested.

The leachability of wool and wool-based geopolymers was checked. Such research has not been conducted so far. Moreover, that is also new, and the article compares the vulnerability of mineral wool and geopolymer made of mineral wool to the leaching of soluble components. The geopolymer was washed out in a comminated form (crushed geopolymer pieces <10 mm). The study allowed us to determine the size and type of harmful substances that may negatively affect soil, groundwater, and surface waters.

The second part of the research investigated the mechanical properties of glass or rock wool-based binder and mortar.

## 2. Materials and Methods

### 2.1. Materials

#### 2.1.1. Mineral Rock and Glass Wool

The materials used for the tests were waste generated in the mineral wool production cycle. Rock wool (Figure 1a) and glass wool (Figure 1b) were tested. 

According to the current Regulation [41], the examined wastes were assigned the 10 12 99 for rock wool and 10 11 03 for glass wool. These are non-hazardous waste. The advantage of the tested waste is its homogeneity and high purity. The manufacturer has introduced waste segregation at the “source” in the production cycle. However, the production cycle cannot use the tested waste due to its form. So far, it has been mainly deposited in landfills for non-hazardous and inert waste and stored in separate quarters due to their character and properties.

The wool density was 10 kg/m³. Mineral wool has a low coefficient of thermal conductivity (so-called lambda, λ). Products made of glass or rock mineral wool, most found on the market, have a thermal conductivity coefficient in the range of 0.030–0.045 W/(m·K). In the next stage of research, the chemical composition was checked. The investigated compositions of tested wools are given in Table 2. The chemical compositions of the rock and glass wool were also determined using the ICP-OES method [7].

#### 2.1.2. The Methodology of Making Wool-Based Geopolymers

The materials used for the tests were glass and rock mineral wool grounded in a ball mill as the target geopolymer powder (Figure 2). The wool was ground for one hour in a Los Angeles drum equipped with additional grinding balls. The density of green wool in the bulk state was 0.935 kg/dm^3^.

In case of an increase of Al in ground glass wool, a bauxite or Al_2_O_3_ power was used. In both cases, Geopolymers were made with ground bauxite or Al_2_O_3_ in 15% of grounded mineral glass wool. The geopolymers from ground rock wool were prepared without adding another component.

Bauxite contains overwhelmingly: hydrous aluminium oxides, aluminium hydroxides, clay minerals, and insoluble materials such as quartz, hematite, magnetite, siderite, and goethite. The aluminium minerals in bauxite can include gibbsite Al(OH)_3_, boehmite AlO(OH), and diaspore, AlO(OH). Table 3 presents the chemical composition of tested bauxite.

The geopolymerisation process requires the dissolution of the starting material in a high pH (alkaline) solution, and thus pH values of fresh geopolymer pastes are usually 11.2–13.2 [37,38]. Sodium oxide reacts exothermically with cold water to produce a sodium hydroxide solution. A concentrated solution of sodium oxide in water will have pH 14. Na_2_O + H_2_O → 2NaOH. When the OH- concentration of the aqueous solution is reduced by ten times, pH is decreased by only one. According to publications [1,3,7,19,24,25,26,27,28,29,30,31,37], NaOH and soda-silicon water glass are adequate alkane sources to achieve the high compressive strength of geopolymers. The study [37] analysed the effect of NaOH concentration (6–14 M) on the mechanical properties of kaolin geopolymers. Compressive strength results showed that the optimum NaOH concentration is 8 M. Moreover, according to research results [3,4,11,12,13,14,16,17,18,19,37,38,39,40,42,43], the soluble glass to 8 M of NaOH solution ratio should be around 2.5 to give geopolymers high compressive strength [37,40]. Therefore, such recommendations were adopted to design the ratio of the alkaline solution in the analysed studies.

For the preparation of paste and mortar was used 450 g of ground wool were and 225 g of alkaline solution were mixed previously in a magnetic stirrer (Figure 3b). The grounded wool to alkali weight ratio was 0.5. The alkaline solution was cooled down to 20 degrees Celsius before ground wool. The paste and mortar were mixed according to the methodology described in EN-196-1 [44]. The value of pH of fresh geopolymers was also tested (Figure 3a).

In the case of geopolymer mortar, normalised sand according to EN-196-1 was used in an amount of 1350 g. 

The slump flow of mortars was measured before and after the table was jolted (according to EN 1015-3 [45]).

The geopolymer paste and mortar was formed as 20 mm × 20 mm × 160 mm (Figure 4a) and 40 mm × 40 mm × 160 mm (Figure 4b) specimens, respectively, and were heat-treated for 48 h at 70 °C. 

The tested geopolymers had been matured in laboratory conditions for 26 days. The air temperature of 20 °C and humidity was about 50%. After that time, they were crushed (Figure 5a,b), then water extracts were made.

The leachability of pollutants from geopolymers produced on the base of waste mineral wool might relate to the form in which they occur, translating into a harmful environmental footprint. In the case of monolithic structures, the leachability level may be determined by the surface release process and diffusion. However, in the case of fragmented forms, the leachability of pollutants determines the percolation process. Therefore, the article also presents.

### 2.2. Research Methodology

#### 2.2.1. Leachability of Mineral Wool

The research procedure included the preparation of water extracts from waste mineral wool (rock and glass), executing chemical tests, and assessing criteria to allow the waste to be stored. The tested waste was mechanically milled for chemical analysis; also, water extracts were carried out on the pre-prepared samples. The samples were tested for the content of organic carbon (TOC)—EN 1484:1999 [46]. Loss on ignition (LOI) was determined according to EN 15935:2013-02 [47]. However, the determination of calorific value by calorimetric bomb combustion to PN-ISO 1928:2020-05 [48]. The procedure for preparing water extracts from solid waste was carried out according to EN 12457-2:2006 standard [49]. Water extracts for waste were prepared at a liquid to solid ratio of L/S = 10 dm^3^/kg (basic test). The prepared water extracts were shaken on a laboratory shaker for 24 h, after which time the obtained extracts were filtered. The leaching liquid was distilled water with a pH of 7.1 and a specific conductivity of 61.18 µS/cm. The analysis of water extracts from waste included several determinations. First, the pH of water extracts was determined using the potentiometric method using the Elmetron CPC-501 apparatus available in Silesian University of Technology in Gliwice, Poland [50]. Chloride (Cl^−^) contents were determined by the Mohr method with the use of silver nitrate (V) as a titration reagent and potassium chromate (VI) as an indicator (PN-ISO 9297:1994) [51]. Determination of sulphates (VI) (SO_4_^2−^) was carried out using the gravimetric method with barium chloride according to PN-ISO 9280:2002 [52]. Determination of phosphorus was according to EN ISO 6878:2006 [53]. The sodium, calcium, potassium, lithium, and barium content in water extracts were determined by flame emission spectrometry—following PN-ISO 9964-3:1994 [54]. Using the GBC AVANTA PM spectrometer available in Silesian University of Technology in Poland, trace element concentrations were determined by flame atomic absorption spectrometry.

#### 2.2.2. The Leachability of the Wool-Based Geopolymer

The execution of water extracts from monolithic and fragmented geopolymers was executed following the EN 12457-4:2006 standard [49]. Monolithic geopolymers were mechanically grounded to grain sizes <10 mm. Water extracts were prepared from the prepared samples with a liquid to solid ratio of L/S = 10 L/kg (basic test). The leaching liquid was distilled water with a pH of 7.1 and a specific conductivity of 62.18 µS/cm. The extracts were then shaken on a laboratory shaker for 24 h, and the suspension was filtered. The analysis of water extracts from fragmented geopolymers covered several determiners. First, the pH of the solutions was carried out using an Elmetron CPC-501 apparatus [50]. Chloride contents were determined by the Mohr method with the use of silver nitrate (V) as a titration reagent and potassium chromate (VI) as an indicator (PN-ISO 9297:1994) [51]. In addition, the determination of sulphates (VI) (SO_4_^2−^) was carried out with a gravimetric method with barium chloride, according to PN-ISO 9280:2002 standard [52]. The flame emission spectrometry method determined sodium, calcium, potassium, and barium content in water extracts from ashes and concrete according to the PN-ISO 9964-3:1994 [54] standard. Using the GBC AVANTA PM spectrometer, trace element concentrations were determined by flame atomic absorption spectrometry.

#### 2.2.3. The Mechanical Properties of the Wool-Based Geopolymer

The geopolymers were matured in laboratory conditions for 26 days. Ambient temperature 20 °C and humidity around 50%. After 28 days, the flexural and compressive strength of the wool-based geopolymers binders and mortars were tested (Figure 6) acc. EN 196-1:2016 [44].

## 3. The Results and Their Discussion

### 3.1. Mineral Wool Leachability Test Results

Table 4 shows the leachability levels of harmful substances that may become an environmental hazard. The analysed mineral wool was slightly alkaline and had a relatively high leachability of chlorides and sulphates.

Since the following values are exceeded: chloride (in both types of wool), sulphates (in glass wool), barium (in glass wool) and cadmium (in rock wool), it can be said that mineral wool does not qualify for storage at a landfill for inert waste. However, the situation is different in the case of non-hazardous and inert waste, which is not municipal waste. Based on the results, it can be concluded that the values obtained using the primary test for Rock wool (the ratio of liquid to solids equal: 10 L/kg) did not exceed the permissible leaching limits. Thus, rock wool can be stored in a landfill for non-hazardous and inert waste. On the other hand, too much barium in a tested sample of glass wool discriminates against it for storage in this landfill. Therefore, in this case, it must be deposited in a hazardous waste landfill to ensure that all results do not exceed the leaching limit values. As a curiosity in the use of waste mineral wool, the effect of using the addition of mineral wool can be cited from crops under cover and municipal sewage sludge on water retention and heavy metals. The average soil’s leaching efficiency (Pb, Zn and Cd) was assessed. Sewage and mineral sludge Wool, widely used in many soil remediation technologies, has been found to have a beneficial and diversified influence on soil water properties and the mobility of heavy metals [56].

Table 5 presents the parameters considered as additional criteria allowing waste to be stored in non-hazardous and inert landfills. Again, a much more significant loss on ignition of glass wool relative to rock wool can be seen. On the other hand, the opposite situation occurs in the case of organic carbon, where rock wool has a much more significant amount (Rock wool contains more organic compounds than glass wool). Therefore, both glass wool and rock wool meet the requirements for waste destined for storage in landfills other than neutral and hazardous.

Determination of the ignition loss at 950 °C is an essential parameter for waste in construction. In addition, it is a fire safety parameter. Loss on ignition at 950 °C temperature for rock wool was at a loss of 4.59%, while it was at 7.88% for glass wool. The rock wool mineral residue had a loose consistency, and it can be stated that it was like sand (Figure 4a). On the other hand, glass wool at the temperature of 950 °C had a liquid consistency; it underwent vitrification (Figure 7b).

As part of the research, the content of volatile elements was determined. The volatile element content for Rock wool was at the level of 3.74%, while about 5.82% for glass wool. When analysing the samples after denoting, carbonisation of the samples was observed (Figure 4c,d). The conducted analyst in thermal transformation confirmed that the material is non-flammable rock wool is characterised by a higher leachability of selected elements than glass wool. 

Only the content of phosphorus was higher for glass wool. The difference can be seen in the lithium content, almost 17 times higher in rock wool (Table 6). All the results met the required conditions regarding the acceptable limit values for waste intended to be stored in landfills. 

### 3.2. The Results of the Study of Leaching Mineral Wool-Based Geopolymer

Similar studies of the leachability of geopolymers made from mineral wool have not been carried out. This is the first study to assess the environmental effects of mineral wool-based geopolymer. The leachability of metals depends on the pH of the material in which they are immobilised. The paper [58] presents the results obtained in the pH leaching test. It assesses the influence of pH changes and occurring processes on releasing heavy metals (Cd, Ni, Cr total, Pb, Cu and Zn) from metallurgical slag in a zinc smelter. Based on test results obtained in the pH test, a strong dependence of heavy metals leaching on the pH was found. The highest concentrations of the analysed elements were observed in an acidic environment. For most metals, except for lead, an increase in the pH of the solution caused a decrease in their concentration. Lead showed an upward trend of release under alkaline conditions. A sharp increase of copper leaching at pH 10.5 was also observed. Based on the study results, cadmium can be considered the most mobile element from metallurgical slag. Chromium indicated the lowest degree of release.

The results of the leachability test (Table 6) were compared to the highest permissible rates of pollution in sewage discharged into water or soil. The water extract of geopolymer binders (SG_B and SG_Al_2_O_3_) was characterised by a strongly alkaline pH value, exceeding the permissible values. The remaining ions are eluting themselves in amounts lower than the limit values. The analysis of the test results presented in Table 7 and Table 8 proves that the geopolymer made of mineral wool meets the requirements in terms of chloride, sulphate, phosphorus, potassium, calcium, lithium, and barium concerning pollution indicators in wastewater discharged into water or soil. Unfortunately, the pH value and sodium content exceed the acceptable environmental standards.

Table 8 summarises the results of studies on the leachability of heavy metals from the geopolymer. The research results on the content of heavy metals in water extracts showed that the rock wool geopolymers in the comminute form meet all the requirements in this regard.

Table 9 shows the leachability of harmful substances from the monolithic geopolymer mortar. The obtained results were related to the leachability of monolithic cement mortars based on Portland cement. The geopolymer mortar was characterised by a high reaction value (pH > 11). The exceedance was also noted for barium, and its content exceeds four times the permissible value. However, the remaining substances do not exceed the allowable values of water and soil.

The leachability of heavy metals from the monolithic form of geopolymers is presented in Table 10. The content of heavy metals in water extracts from geopolymer mortar did not exceed the permissible values. Therefore, it can be said that they do not pose a threat to the natural environment.

The rock wool and glass wool show different leachability performances for each ion because their base composition is very different [7]. Table 11 The chemical composition of rock wool and glass wool, determined by ICP-OES.

The presented test results proved that it is possible to store rock wool in a landfill for non-hazardous and inert waste. However, too much barium in glass wool discriminates against it for storage in such landfills; therefore, it should be stored in a hazardous waste landfill. Moreover, the geopolymer is characterised by too high alkalinity due to the environmental requirements.

The proposed reuse of waste mineral wool in the form of a geopolymer is a solution that is beneficial for the environment, climate, human health, and economic reasons.

The results of the research on the quality of water extracts made of comminute geopolymer binders based on rock wool show that:the pH is at the level of 12.2,the content of sulphates (SO_4_^2−^) is present in amounts from 298 to 342 mg/L, depending on the added recycling material,sodium content was at the level of 2700 mg/L.

On the other hand, geopolymer products are used in the industry, such as PCI-Geofug geopolymer grout by Basf [59], GeoLite geopolymer mortar by Kerakoll [60], ASTRA GKB geopolymer concrete [61], also the geopolymer injection as a quick and non-invasive method of strengthening the ground. It is used in linear (infrastructure) and cubature construction (industrial, commercial, and residential). It has been successfully used in Western Europe and Scandinavia for over 40 years. Geopolymer is gaining popularity due to its exceptional performance parameters and convenience of use: speedy repair time and minimal nuisance during geoengineering works. Unfortunately, the results of the leachability test of hazardous substances and heavy metals do not pose a potential threat to the environment. However, a strongly alkaline reaction and the excess sodium content are open for further consideration. The authors will lower the alkaline reaction of geopolymer binders in further research steps. The geopolymerisation process requires the dissolution of the starting material in a high pH (alkaline) solution, and thus, pH values of fresh geopolymer pastes are usually 11.2–13.2 [37,38]. Like products based on Portland cement, the pH must not be too low due to reinforcement corrosions [55]. The concrete is acceptable for use, although the pH of the water extract is from 9 to 11 (Table 9). Due to environmental requirements, the value of the concrete solution is also too high, but not as high as in the case of the geopolymer solution. The authors do not know the pH of the mentioned above geopolymer products. The authors want to check out it is possible to lower the pH geopolymer add an acid component to it, but this will drastically reduce its strength. The authors will analyse this problem in the future. The problem is very significant in terms of the environment. In the future, the authors will decrease the alkalinity of wool-based geopolymers to 11.2 according to the suggestion of publication [25,42].

### 3.3. The Fresh and Mechanical Results of Wool-Based Geopolymer

The slump-flow of glass wool-based geopolymer fresh mortar with Al_2_O_3_ or bauxite was comparable and was about 13 cm (Figure 8). Whereas the slump flow of geopolymer mortar with rock wool was more significant by up to 3 cm. thus, the type of wool is essential for the rheological properties of fresh geopolymer. It has been noticed that geopolymers, due to their high viscosity value, are very resistant to segregation, which is undoubtedly their great advantage.

The geopolymer was characterised by high adhesion strength to the forms in which they stayed. The phenomenon of geopolymer adhesion was mentioned in the publication [18]. It was noticed that geopolymers show more adhesion than materials at the entrance to cement. This phenomenon may be due to work with the fibres or reinforcement of the geopolymer. Moreover, the geopolymers were highly viscous, especially with sodium-potassium water glass, which requires careful mechanical concentration. In Figure 9, pores in the structure of wool-based geopolymers are presented, probably from under dumping the sample. Despite this, geopolymers glass wool-based mortars with Al_2_O_3_ (Figure 9) obtained an average compressive strength of 59 MPa and tensile strength of 4.5 MPa. The geopolymer with bauxite achieved about 51 MPa, and flexural strength 4.1 MPa. Thus, Al_2_O_3_ is a better additional glass wool-based geopolymer than bauxite. 

The average compressive strength of their binders (tested on 20 mm × 20 mm × 160 mm samples, Figure 10) was about 20 MPa (in the case of glass wool-based geopolymer with bauxite). It was observed that samples of geopolymeric binders without aggregate participation are characterised by cracking and deformation of samples due to shrinkage. The reason was that the samples were too slender. The shrinkage value is greatly influenced by the geometry of the samples, especially without aggregate. The compressive strength of binders with AL_2_O_3_ was about 25 MPa.

The average compressive strength of rock wool-based geopolymer mortar was about 62 MPa, and their compressive strength of binder was about 25 MPa and flexural strength 4.8 MPa. It was also noticed that the rock wool thickens better in the moulds, which translated into a higher value of their strength.

The studies [6,7] show that maximum compressive strengths of 48.7 and 30.0 MPa were measured for wool-based binders. The binder matrix consisted of aluminosilicate gel with partly dissolved mineral wool fibres. The maximum flexural strength was 13.2 MPa for GW and 20.1 MPa for RW. This study shows that high strength can be obtained without additional co-binders by activating alkali with sodium aluminate solution. Furthermore, the research carried out by the authors of this publication proved that it is possible to obtain the strength of the geopolymer, resistant to shrinkage deformation, but after modification of its composition. The latest achievement of the authors is a wool-based geopolymer with a strength of 100 MPa, which will be the subject of the following publication.

## 4. Conclusions

Due to the increasing necessity to reduce CO_2_ emissions and the energy consumption of buildings, the consumption of mineral wool in construction is increasing. Mineral wool is a waste material that meets the requirements for storage; however, it is not the direction to be followed when considering the future. In line with the closed-loop policy and the increasing need to reuse materials, the goal is to recycle them, also due to the decreasing availability of the waste storage area.

The carried-out research results proved that:It is possible to store rock wool in a landfill for non-hazardous and inert waste. However, in this case, too much barium in a glass wool sample discriminates against it for storage in a non-hazardous landfill; storing it in a hazardous waste landfill seems necessary. Leachability tests were performed to investigate the environmental effects of the wool-based monolith and crushed forms of wool-based geopolymers. The geopolymer was washed out in a comminated form (crushed geopolymer pieces <10 mm). The study allowed us to determine the size and type of harmful substances that may negatively affect soil, groundwater, and surface waters.The proposed use of wool for geopolymer binders is a correct solution for environmental reasons. The geopolymer meets the guidelines of the Regulation of the Minister of Maritime Economy and Inland Navigation of 12 July 2019, on substances particularly harmful to the aquatic environment and the conditions to be met when discharging sewage into waters or soil. As well as during the discharge of rainwater or meltwater into waters or water facilities (Journal of Laws 2019, item 1311), except for the pH, the maximum permissible value of which should be 9.0, and the sodium content, the ultimate value of which should be 800.Due to their high viscosity value, the wool-based geopolymers are resistant to segregation, which is undoubtedly their great advantage. Moreover, the geopolymer is characterised by high strength of adhesion. Therefore, the type of wool is essential for the rheological properties of fresh geopolymer. Moreover, the geopolymers were characterised by high viscosity, requiring careful mechanical concentration.Geopolymers glass wool-based mortars with Al_2_O_3_ obtained an average compressive strength of 59, the geopolymer with bauxite achieved about 51 MPa, and flexural strength 4.1 MPa. Thus, Al_2_O_3_ is a better additional glass wool-based geopolymer than bauxite. Moreover, Al_2_O_3_ may also be used as a waste material, which contributes to the eco-friendliness of the geopolymer.The average compressive strength of rock wool-based geopolymer binder was about 20 MPa. It was observed that samples of geopolymers grout without aggregate participation are characterised by cracking and deformation. The average compressive strength of rock wool-based geopolymer mortar was about 62 MPa, their binder’s compressive strength was about 25 MPa, and flexural strength was 4.8 MPa.

The compressive strength of geopolymer results [37] showed that the optimum NaOH concentration is 8 M. The geopolymer strength decreases with NaO concentration in the NaOH solution. As mentioned in the article, the geopolymerisation process requires the dissolution of the starting material in a high pH (alkaline) solution. Thus, the pH values of fresh geopolymer pastes are usually 11.2–13.2 [37,38]. Therefore, an attempt will be made to lower the wool-based geopolymer alkalinity to 11.2. Although it still does not meet the environmental requirements [1,57]. However, the problem needs future research.

In the following research steps, Al_2_O_3_ will be replaced with a waste material that predominantly contains the same compound to create a glass wool geopolymer from waste materials only. 

Further modifications aim to replace the sand of geopolymer mortar with material derived from waste, in line with the goals of sustainable development and the protection of natural resources.

## Data Availability

Not applicable.

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
