# Peer review of "The Comparison of the Environmental Impact of Waste Mineral Wool and Mineral in Wool-Based Geopolymer"

_materials, 2022, doi:10.3390/ma15062050_

Round 1

Reviewer 1 Report

Please, in the Title part, add – in wool-based.

General comment: The paper’s topic is interesting and addresses a very important aspect of environmental protection: using waste construction materials again, especially glass and stone wool.

Also, authors must improve the manuscript with up-to-date references, it is a mandatory request. The paper is scientifically poor written it demands major revision.

Abstract: Line 16: Please rephrase the second sentence it is unclear.

Line 17,18: Please merge these two sentences in one more precise.

Line 18: It should be written like this, it is a suggestion: “Waste mineral wool from demolition sites, belongs to the specific group of waste, where the storage and collection require implementation of restrictive conditions, like improper storage of mineral wool which is highly hazardous for the environment.”

Line 21: Please use proper symbols uppercases and low cases for Cl- anion, and SO42- it is typo mistakes that are not allowed for one scientific paper. Also, in line 25 for Al2O3!

Line 35,36 please merge sentences in one.

Line 36: The sentence should be deleted or rephrased it is unnecessary.

Line 49-64: It is a scientific paper regarding materials science not a regulatory of waste management. It is ok to emphasize waste management importance but not the whole paragraph. The 3 or 4 sentences are enough. In my opinion, this part should be rephrased.

Line 66-81: REFERENCES in-text?!?!

Line 79-83 and 84-93: This part regards geopolymers, and there are a lot of references in literature that can be found, so please add references in the text.

Line 106-125: The text seems well written, but references are mandatory. Line 110: Geopolymers based on what? Clays or fly ash as starting material can be used as a concrete replacement? Please delete “WHAT” at the beginning of the sentence and rephrase it.

Figures 1,2,3 and 4 can be merged into one figure. Please merge it and rephrase the text according to changes.

Line 174: Please add a reference at the end of the first sentence.

Figure 6 is unnecessary because no significant difference is seen between Figures 1 and 2 and this figure. Also, loss of ignition can be seen from results and text.

Figures 7 and 8 also can be merged.

The discussion part is written like a conclusion. It must be reorganized and also changed references and a comparison of the obtained results with the literature data must be written.

The conclusion part should also be rewritten. The study’s main results should be emphasized with an indication and emphasis on why and in what way it is good to use newly obtained geopolymer materials. This study has not been completed, and future research should be considered.

Author Response

Thank you very much for your comments. The suggestions were included in the modified article.

Reviewer 2 Report

  1. Abstract: Row no. 15 to 19 to be moved to the introduction part or removed. avoid General things and it should be specific to your study.
  2.    Row no. 21, what is SO42 ? use Subscript or create small letters below the text line.
  3. Row no. 26, Al2O3 or Al2O3
  4. Introduction Row 39 to 42, for example,123 million Mg (tonnes) of waste -should be cited properly.
  5. Materials and Methods( 2.1.1. Mineral rock and glass wool) chemical properties of mineral and glass wool to be included and also check the whether the material is rich in silica and alumina.
  6. 2.1.2. Geopolymer made of glass mineral wool- Row no.151 mix proportion of geopolymer to be included.
  7. Fig 3 & 4 the measurement scale  (1 Cm to 15 Cm ) what it indicates?
  8. 3. Results - this part is to be improved also clearly justify your results with some citation.
  9. 4. Discussion - highlight your own results, not for others.
  10. Row 319 to 366 should be removed. kindly include your experimental results and discussion.

Author Response

(The authors gave the same response as above.)

Reviewer 3 Report

The manuscript presented a well-organized research paper. The paper is very interesting and worth publication in the Materials in the reviewer's view. However, following comments are advised to be considered before acceptance:

Abstract

The important findings obtained from the experimental studies can be given in this section with a sentence.

Page 1; Line 26: Al2O3 should be written with subscript. (Al2O3)

Introduction

The introduction needs to be more emphasized on the research work with a detailed explanation of the whole process considering past, present and future scope. How the present study gives more accurate results than previous studies? It needs to be strengthened in terms of recent research in this area with possible research gaps. The literature can be improved with recent studies in this field. In addition, the type of rocks for mineral wool production and various activators used for geopolymers can be mentioned here. Do you have any data about the rock and glass wool waste amount from construction demolition?

Materials and Methods

What are the types of mineral wool and glass wool? Were they obtained from a demolition area or another source? Please specify this here. If possible physical properties of these materials can be given.

Page 4; Line 150: 1.41 g / cm3 can be written as 1.41 g / cm3 (superscript)

Did you investigate the physical and mechanical properties of the produced prismatic geopolymer specimens? (Unit weight compressive strength etc.)

Results

Page 8; Line 263: Please write mg/dm3 with superscript as (mg/dm3)

Discussion

Page 11; Line 319 to 347: This part can be given in the Introduction part depending on the general knowledge about the geopolymers.

Conclusion

The benefits of geopolymers produced by rock and glass wool against cementitious composites can be emphasized in this section. 

Please mention your study limits and suggest some future research topics.
The authors are advised to write the conclusion in a comprehensive way, it should contain key values, suitability of the applied method, contributions and possible future work.

Author Response

(The authors gave the same response as above.)

Reviewer 4 Report

This work describes an interesting study in using wools as alkaline activated geopolymers for construction materials. The leaching tests were performed attending to European standards, revealing no concerns regarding the presence of metals but showed high pH levels. The authors addressed well the main aim of the work. However, some concerns should be clarified by the authors.

1. In a general way, the English should be improved through the text, especially in the Abstract and Conclusion sections.

2. Lines 21, 26, 150, 188, 226: please change the following anions and units attending to their superscript and/or subscript accordingly, as: Cl-, SO42-, 1.41 g/cm3, SO42-, dm3.

3. The units for the amount of waste generated through the full manuscript should be clarified. For example, in line 40 is written: “123 million Mg (tonnes) of waste”. I suppose that the authors intend to say megagram (Mg), but megagram is not commonly used. In fact, Mg is often related with the chemical element for magnesium, which can confuse the readers. I was not able to check if the units provided in the lines 40 – 42 were correct because no reference was provided. Please, provide references for these values. This issue is repeated in lines 112 and 113. I strongly suggest changing all Mg units in the manuscript to millions of tons or similar.

4. Some adjustments in the references present in the text should be made:
Line 83: “…rocks. [3]”. It should be “…rocks [3].”

Line 325: “…1800 mm. and also…”. It should be “…1800 mm and also..”. In fact, 1800 mm looks strange, why not 1.8 m?

Line 341: “…Portland cement. [26] So far,…”. It should be “…Portland cement [26]. So far,…”.

Line 353: “…mineral additives. [31].” It should be “…mineral additives [31].”

5. Bauxite and Al2O3 were added to the wool waste for the geopolymer preparation. The mentioned bauxite is the bauxite ore? If so, the authors can provide a general chemical characterisation of the ore? Where did the authors obtained/purchased bauxite from? The name of the company should be mentioned in the Materials and Methods section and more details should be added about bauxite. Moreover, there is no chemical composition related to the wools provided (content of CaO, MgO, SiO2,….). Since the authors are submitting the manuscript to a Material Science based journal, I believe that more information about the chemical phase composition of the raw materials used should be provided.

6. Line 149: “…8 molar NaOH solution…”. The molar unit is M, so please change to “… 8 M of NaOH solution …”.

7. The name of the companies were all the reagents were purchased should be also provided in the Materials and Methods section along with the name of the chemical reagents and their purity grade.

8. What the authors intend to say by dm3/kg (basic test)? It is dry basis? Can the authors, please clarify in the text? (Lines 181, 200). The dm3/kg should be written without space before and after the /. About the dm3, the SI unit for volume is L instead of dm3.

9. The name of some elements present in the Tables are not properly written in English.

Table 1: Zink, Zn and Chrome, Cr. It should be Zinc and Chromium, respectively.

Table 3: Lit, Li. It should be Lithium.

Table 4: General phosphorus, P. Do you mean total phosphorus? What about the P in Table 3 and 6? Chlorides and sulphates should be written as Cl- and SO42-.

Table 5: Chrom, Cr. It should be Chromium.
Table 6: Lit, Li and Bar, Ba. It should be Lithium and Barium. Chlorides and sulphates should be written as Cl- and SO42-.

Table 7. Chromium.

10. Line 240, 241, 257: Sometimes °C appears with an extra space (° C) and others as ºC. It should appear only as °

11. Line 371: please remove “the concentration of hydrogen ions (pH) is...”. “The pH is at the level of 12.2” is already suitable.

12. The geopolymers obtained revealed to have a high pH after the leaching tests, most likely due to the high concentration of the NaOH used during its activation. Why did the authors used 8 M of NaOH? There are no references about other works in the text that explain the decision, but which is usually the concentration of NaOH used for the alkaline activation of these types of geopolymers in literature?

13. Despite the high pH value obtained in the leaching tests, the geopolymers seem to have suitable properties to be used in construction materials. Attending to the polish legislation, which is the highest pH level for accepting geopolymers for building construction materials? There is any legal limit? Can the authors exemplify in the text, which constructing materials can be used for the geopolymers incorporation?

14. The reference list is not in agreement with the reference formats attending to books, papers,.., usually required when publishing in MDPI journals. The authors should check it carefully attending to the MDPI guide, which is available on the website.

Author Response

(The authors gave the same response as above.)

Reviewer 5 Report

The manuscript discusses the processing of mineral wool into geopolymer as a potential solution to the mineral wool waste issue. The manuscript is well written, except at several locations where improvements may be required. However, articles lacks significant information at multiple fronts and needed to be incorporated for the better understanding among the readers. The questions, suggestions and concerns are as follows:

  1. In the Introduction section, it is not clear why authors selected the bauxite and Al2O3 for this study?
  2. In the introduction section, on page 3, Line 110-116 are comprising a lot of expectations from the geopolymer. However, the insulating characteristics, protection against moisture and resistance towards fungi growth formation has not been discussed in the article. I suggest authors to keep the introduction focused on the research target.
  3. On Page 1, authors mention ton as tonnes, while on page 3 the notation becomes ton. Kindly unify. Furthermore, author may choose one of the unit for exemplifying the waste or amount i.e. either Mg or ton.
  4. Kindly explain section 2.1.2 in more detail. The reason behind the selection of this procedure is unclear. Kindly add the appropriate references or the explanation in support of the same. 
  5. In all the Tables across the article, no error bar or standard deviation for the data has been mentioned. I suggest authors to incorporate the standard deviation/error bars. It is helpful to understand the results reproducibility and reliability. 
  6. In Table 1, why cadmium was able to leach through the rock wool but unable to do so in the case of glass wool?
  7. In similar lines, in Table 2, Why rock wool and glass wool are showing different leachability performance for each of these ions. In the absence of the reason behind this performance, it is difficult to understand that whether the adopted research design is appropriate or not.
  8. In Table 3, does % sm represents % supplementary materials? 
  9. In Table 4, does author believe that if the geopolymer synthesis was performed with lower alkaline concentrated solution the performance towards the pH may come in range? Kindly comment.
  10. In Table 5, please unify the performance for the synthesized geopolymer with the tested rock wool and glass wool. For example, in case of Pb: In Table 1 authors mention <0.30; while in Table 5 authors mention for Pb leachability as 0.22. In such inconsistency, kindly mention both data as <0.30 or give numbers for both of them for ease of comparison and understanding. 
  11. Kindly improve the discussion section of the manuscript as suggested above by including the reasons (or plausible reasons) for the observed results. 

Author Response

(The authors gave the same response as above.)

Round 2

Reviewer 1 Report

The authors have changed manuscript according to suggestions. 

Author Response

The changes have been made.
The authors have changed manuscript according to suggestions. 

Reviewer 2 Report

  1. Abstract-  crisp statement can be included .

  2. 2. Materials and Methods-Row 231 table 2 Description - Fe2o3 "O' means oxygen not zero. it should be modified.

Author Response

The authors have changed manuscript according to suggestions. 

Reviewer 5 Report

Authors made an attempt to reply the concerns and questions of the reviewer. However, the crucial aspect of this research in terms of research novelty and material selection is not satisfactorily responded. The unanswered questions/suggestions in this manuscript continued to be the following. I would like to request authors to kindly consider the suggestions and answer the concerns satisfactorily for the better understanding among the readers.

  1. The added sentences in the Introduction on page 3 are unreadable due to weak contrast in the color selection (yellow font over the white page) of the font. Kindly improve the text color to make it readable for the reviewers.
  2. In the response to the Question #6 and #7, authors have cited literature. However, the question was not whether someone has performed similar experiment or not but rather on the performance of the samples under consideration. Kindly answer the questions satisfactorily. 
  3. In the response to the Question #4 regarding the reason for the current method of preparation selection was not found. Kindly mention the newly added concerned sentences in the response letter as well for easier and faster navigation. 
  4. On page 7, authors mention that the pH value of NaOH depends on the amount of Na in the NaOH solution. Kindly explain why the NaOH solution has Na metal without being oxidized? Secondly, pH is dependent on the hydronium ion concentration, how does author correlate it with the Na concentration?
  5.  In Table 5, the limit values have been reported as a whole numbers, however for the Rock wool and Glass wool authors prefer the accuracy up to two decimals. Is it necessary?
  6. In Table 8, 10 please correct element name as a Chromium. 

Author Response

(The authors gave the same response as above.)
